# Effect of aging and body characteristics on facial sexual dimorphism in the Caucasian Population

Zala Skomina[1]*, Miha Verdenik[2], Nataša Ihan Hren[1,2]

1 Department of Maxillofacial and Oral Surgery, Faculty of Medicine Ljubljana, University of Ljubljana, Ljubljana, Slovenia, 2 Department of Maxillofacial and Oral Surgery, University Medical Centre Ljubljana, Ljubljana, Slovenia

* zala.skomina@kclj.si

**Data Availability Statement:** All relevant data are within the paper and its Supporting Information files.

**Funding:** The authors received no specific funding for this work.

## Abstract

### Aim

The aim of this study was to quantify gender-specific facial characteristics in younger and older adults and to determine how aging and body characteristics, such as height and body-mass index (BMI), influence facial sexual dimorphism.

### Methods

The cohort study included 90 younger adults of Caucasian origin (average age of 45 females 23.2 ± 1.9 and 45 males 23.7 ± 2.4 years) and 90 older adults (average age of 49 females 78.1 ± 8.1 and 41 males 74.5 ± 7.7 years). Three-dimensional facial scans were performed with an Artec MHT 3D scanner. The data were analyzed using the software package Rapidform®. The parameters to evaluate facial symmetry, height, width, profile, facial shape, nose, eyes and mouth characteristics were determined based on 39 facial landmarks. Student's *t*-test was used to calculate the statistical differences between the genders in the younger and older adults and a multiple-linear-regression analysis was used to evaluate the impact of gender, age, body-mass index and body height.

### Results

We found that the female faces were more symmetrical than the male faces, and this was statistically significant in the older adults. The female facial shape was more rounded and their faces were smaller, after normalizing for body size. The males had wider mouths, longer upper lips, larger noses and more prominent lower foreheads. Surprisingly, we found that all the gender-dependent characteristics were even more pronounced in the older adults. Increased facial asymmetry, decreased facial convexity, increased forehead angle, narrower vermilions and longer inter-eye distances occurred in both genders during aging. An increased BMI was associated with wider faces, more concave facial profiles and wider noses, while greater body height correlated with increased facial heights and wider mouths.

**Competing interests:** The authors have declared that no competing interests exist.

## Conclusion

Facial sexual dimorphism was confirmed by multiple parameters in our study, while the differences between the genders were more pronounced in the older adults.

## Introduction

Sexual dimorphism relates to the recognition of two sexes per species and the phenotypic expression of multi-factorial differences at the chromosomal, gonadal, hormonal and behavioral levels [1]. These differences also have the evolutionary significance, and might be adaptations for mate choice [2].

There are known gender differences in facial characteristics. The majority of facial features containing secondary sexual traits develop or increase in size at puberty under the influence of sex hormones. For example, males have more pronounced noses, brows and frontal regions, more prominent chins and larger jaws compared with females [3]. Some studies suggest that women have bigger eyes, smaller noses and thinner lips [4].

The perception of facial attractiveness is, among other factors, influenced by facial symmetry. Symmetry, sexual dimorphism and averageness are good candidates for biologically based standards of beauty [2]. Average faces follow average trait values for a specific population. Averageness is conditioned not only racially, but also ethnically within the race [5]. The symmetry is more pronounced in females, because beauty has a larger role in the male evolutionary principles of female mate selection [6]. Some studies found a positive correlation [7] between masculinity and symmetry in male faces, while others failed to confirm these findings [8].

One of the factors we need to consider in facial sexual dimorphism is aging. A face changes throughout a lifetime and some of the consequences of aging are already known. Facial aging represents the transition from youth, where there is an optimal relationship between bone morphology and the volume of the soft-tissue envelope, to the imbalance between these components that leads to the appearance of an aged face [9]. Facial aging results from a combination of changes in soft tissue (such as changes in the status of elastin and collagen fibers), with bone loss in specific areas of the facial skeleton contributing to the features of aging [10].

Facial appearance has a very important influence on our psycho-social wellbeing. Thus, the appreciation of the characteristics of human faces is important not only in aesthetic surgery but also in craniofacial surgery, especially in orthognathic and syndromic patients, because normal gender differences impact on the planned facial appearance.

The present study aimed to quantify gender differences in the facial characteristics of younger and older adults of Caucasian ancestry in Slovenia. We used noninvasive digital three-dimensional (3D) technology, and in addition to the standard anthropometric analysis of facial parameters, we also quantified facial asymmetry using a novel method of 3D scanning. Our goal was to determine how different body characteristics, such as body height, body-mass index (BMI) and age, influence the facial gender differences in our sample.

## Materials and methods

### Study group

The cohort study included 100 younger adults (50 females, average age 23.2 ± 1.8 years and 50 males, average age 23.6 ± 2.4 years) and 100 older adults (50 females, average age 77.9 ± 8.6 years and 50 males, average age 75.3 ± 7.8 years). The younger adults were students at the

School of Medicine of Ljubljana, Slovenia. The older group contained residents of five retirement homes in Ljubljana, Slovenia. Only individuals of Caucasian origin were included. The exclusion criteria were a craniofacial anomaly, a history of major facial trauma, or orthognathic surgery, facial paresis and tremor. Male subjects with facial hair were also excluded.

The sex, age, BMI, body weight and height of the subjects enrolled in the study are presented in Table 1.

Ethical approval for this study was obtained from the Slovenian National Ethics Committee and written informed consent was obtained from all the subjects.

## Protocol

All the subjects had a 3D facial scan. During the acquisition, special attention was given to positioning the subject and relaxing the facial musculature. Each subject was placed in a clinically reproducible natural head posture, the mandible was in the rest position; they were asked not to swallow, relax the lips and keep both eyes open during the scan. The natural head position was achieved after instructions and exercises by moving the head up and down a few times and then stopping the movement and looking into the distance. A relaxed, closed-mouth position was achieved with a repeated wide opening and closing the mouth until light contact of the lips was achieved A single facial scan required less than 10 seconds, so the subjects were able to maintain their positions.

Surface facial images were obtained using an Artec MHT 3D scanner (Artec Ventures Ltd.), which uses the flying triangulation method to capture a 3D surface. The distance between the examined person and the scanner was 50–70 cm.

The 3D surface was then processed using Artec Studio software to obtain 3D scans in the STL format. Each scan of the face was processed in order to remove unwanted data, bounded by the exterior border beyond the hairline on the forehead, and around the lower jaw angle forward to the sub-mental region under the hyoid bone. A further analysis was conducted using the software package Rapidform®2006 (Inus Technology Inc., Seoul, Korea). Thirty-nine superficial facial landmarks were manually determined on each of the 3D facial scans by a single operator. Before the study, the intra-rater reliability was verified with an intraclass correlation and we confirmed that the method is reliable and that it does not introduce any bias. Based on the facial landmarks, the parameters described below were determined.

**Facial symmetry.** Facial symmetry was evaluated with the 3D mirroring approach. For each subject a mirror facial shell was created using Rapidform®2006. The best-fit superimposition method was used to merge the original and the mirrored shells, as shown in Fig 1. The surface matching between the two shells with 0.5 mm of tolerance was expressed as a percentage. The average distances and the maximum distances between the two shells were also computed.

**Table 1. Basic descriptive statistics of the study sample; number (n), average age in years with standard deviation (SD), average body-mass index (BMI), average body weight in kilograms (kg) and average body height in meters (m) for both genders.**

|  | n | Age (years) | | | BMI | | | Body weight (kg) | | | Body height (m) | | |
|---|---|---|---|---|---|---|---|---|---|---|---|---|---|
|  |  | mean | SD | (Q1) median Q3 | mean | SD | (Q1) median (Q3) | mean | SD | (Q1) median (Q3) | mean | SD | (Q1) median (Q3) |
| Young female | 50 | 23.2 | 1.8 | (21.8) 23.2 (24.0) | 20.9 | 1.9 | (19.5) 20.4 (22.3) | 60.5 | 7.4 | (53.8) 61.0 (65.3) | 1.7 | 0.1 | (1.65) 1.70 (1.76) |
| Young male | 50 | 23.6 | 2.4 | (21.8) 23.6 (25.5) | 23.4 | 2.9 | (21.3) 22.8 (25.0) | 78.0 | 10.6 | (69.5) 78.5 (83.3) | 1.8 | 0.1 | (1.78) 1.83 (1.86) |
| Older female | 50 | 77.9 | 8.6 | (69.2) 78.5 (84.2) | 26.6 | 4.3 | (23.5) 25.4 (28.8) | 70.0 | 13.8 | (60.0) 70.0 (78.0) | 1.6 | 0.1 | (1.57) 1.60 (1.65) |
| Older male | 50 | 75.3 | 7.8 | (68.2) 73.9 (80.0) | 27.4 | 3.7 | (25.2) 27.2 (29.3) | 81.1 | 12.6 | (73.3) 81.0 (88.0) | 1.7 | 0.1 | (1.67) 1.72 (1.78) |

Q1—first quartile; Q3—third quartile

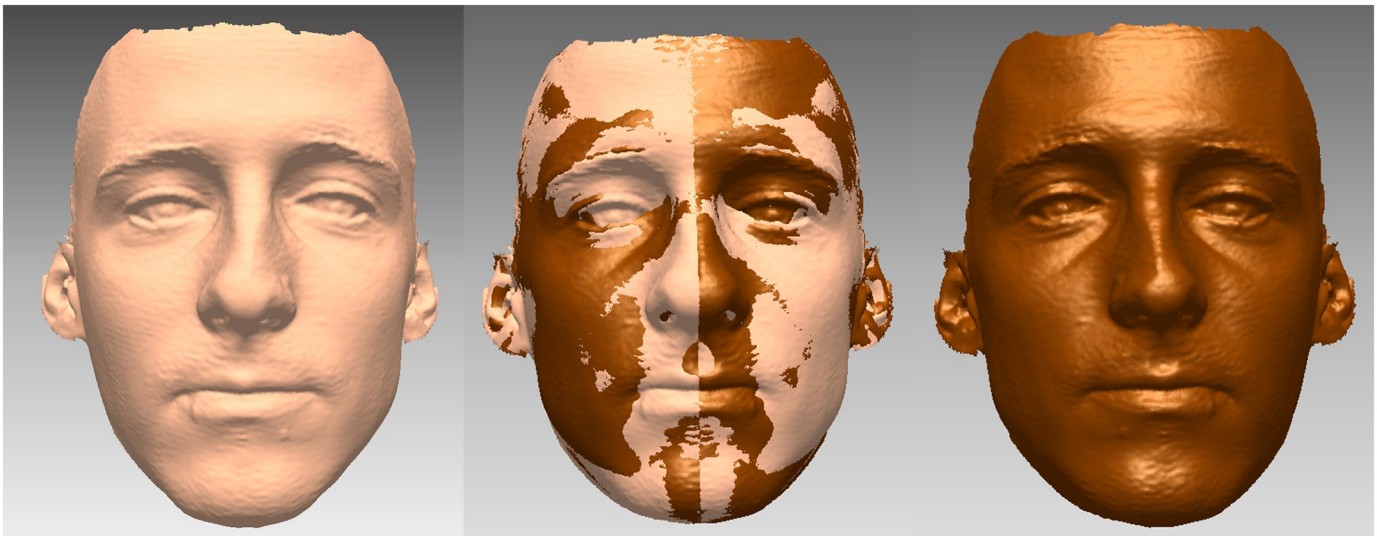

**Fig 1. Facial symmetry was evaluated with the best-fit superimposition method for the original facial shell (left) and the mirrored facial shell (right).** The merged shells are seen in the center.

**Facial widths.**    The upper facial width was defined as the distance between the left and right zygoma. The lower facial width was the distance between the left and right gonion. The ratio between the upper and lower facial widths was calculated to describe the shape of the face.

**Facial heights.**    Several parameters were used to evaluate the height of the face. Facial height was determined as the distance between the nasion and gnathion points. The trichion point (the point between the forehead and the scalp) was not used because it is the most variable point, as a result of hair loss during aging. The middle facial height was the distance between the glabella and subnasale points. The lower facial height was the distance between the subnasale and pogonion points. The ratio between the middle and lower facial heights was calculated to describe which facial part contributes to the facial height changes during aging.

**The ratio between facial width and height.**    The width-to-height ratio was a parameter used to describe the shape of the whole face.

The facial width and height parameters are shown in Fig 2A.

**Facial profile.**    The facial angle was the angle between the the nasion, subnasale and pogonion points. A larger angle means a more concave facial profile. The angle of the lower facial height was the angle between the subnasale, stomion and pogonion points. It describes the facial profile in the lower facial height.

The forehead angle was the parameter used to describe inclination of the forehead. It was the angle between face vertical (the line between the nasion and point a—the most posterior point of the philtrum) and the line between the glabella and trichion.

The glabella's prominence angle was the angle between the nasion, glabella and trichion points.

The facial profile parameters are shown in Fig 2B.

**Mouth.**    Several parameters were used to evaluate the characteristics of the mouth. Mouth width was determined as the distance between the left and right cheilion (the point at each labial commisure). The upper vermilion middle height (the distance between the labiale superior and the stomion) and the lower vermilion middle height (the distance between the stomion and the labiale inferior) were the parameters used to describe the size of the lips. The

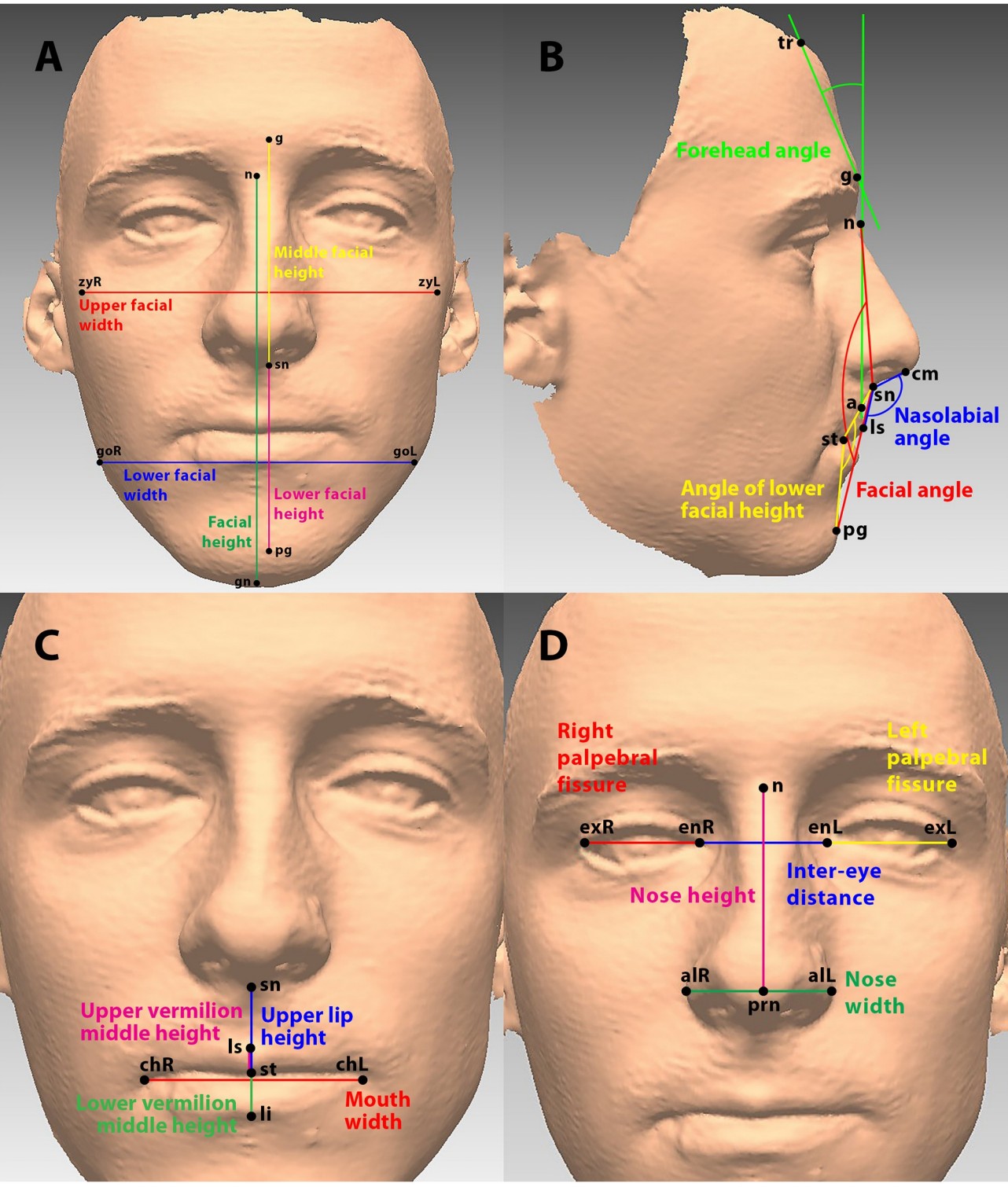

**Fig 2. Facial parameters.** A: Upper facial width (red), Lower facial width (blue), Facial height (green), Middle facial height (yellow), Lower facial height (purple); B: Facial angle (red), Angle of lower facial height (yellow), Forehead angle (green), Nasolabial angle (blue) C: Mouth width (red), Upper-lip height (blue), Upper vermilion middle height (purple), Lower vermilion middle height (green), D: Right palpebral fissure width (red), Left palpebral fissure width (yellow), Inter-eye distance (blue), Nose height (purple), Nose width (green).

upper-lip height was the distance between the subnasale and stomion points. The ratio between the upper-lip height and the lower facial height was also calculated. The mouth parameters are shown in Fig 2C.

**Nose.** The characteristics of the nose were evaluated from the nose height, width, and the angle between the nose and the upper lip (nasolabial angle). The nose height was the distance between the base of the nose (nasion point) and the tip of the nose (pronasale point). The distance between the left and right alae nasi points was the nose width. The nasolabial angle (Fig 2B) was the angle between the upper lip and the tangent on the nose columella.

**Eyes.** The size of the eyes was described with the left and right palpebral fissure width (the distance between the endocanthion and exocanthion points). The inter-eye distance was the distance between the left and right endocanthion. The parameters of the nose and eyes are presented in Fig 2D.

## Statistical analysis

The Statistical Package for Social Sciences 17.0 (SPSS Inc., Chicago, Illinois, USA) was used for the statistical analysis. The data were tested for a normal distribution. The unpaired Student's *t*-test was used to calculate the statistical differences of the parameters between men and women separately for the younger and older adults. A multiple-linear-regression model was used to evaluate the impact of sex, age, BMI and body height on the facial parameters with respect to all the subjects together. The significant regression coefficient (marked with *; **; ***) shows how the dependent variable is expected to change when that independent variable increases by one, holding all the other independent variables constant. For example; in older group face width is 7 mm longer than in younger group with unchanged BMI, body height and gender. Differences were considered to be statistically significant at values of $p < 0.05$.

## Results

The analysis of the facial parameters and the comparison of the genders in the younger adults are presented in Table 2. The influence of aging on sexual dimorphism is presented in Table 3. The results of the multiple linear regression to evaluate the impact of gender, age, BMI and body height on the facial parameters are shown in Table 4

### Facial symmetry

We found that the percentage of surface matching between two shells, a measure of the facial symmetry, was lower in younger males (50.2 ± 10.9%) than in younger females (53.7 ± 9.4%). In the older group the percentage of surface matching between the two shells was 39.2 ± 9.01% in males and 42.1 ± 8.9 in females. Thus, the women had more symmetric faces than the men, with the differences being statistically significant in the older group (Table 3). The average distance between the original facial scans and the mirrored facial scans was larger in the male group. In the group of younger females, the average distance was 0.67 ± 0.16 mm, in the younger male group it was 0.74 ± 0.21 mm. In the group of older people the average distances were larger (older women had 0.85 ± 0.20 mm and older men had 0.97 ± 0.24 mm).

### Facial widths

We found that men had wider faces than women (p = 0.026). On average the men's upper facial width was 3 mm wider and the lower facial width 8 mm wider than the women's (Table 2). With increasing BMI the facial widths increased for both genders; however, body height had no impact on the facial widths (Table 4). The older adults had wider faces than the

**Table 2. Descriptive statistics and comparison of the facial parameters between genders (independent samples *t*-test) in the younger group (45 females, 45 males).**

| PARAMETERS | FEMALES MEAN (SD) | MALES MEAN (SD) | MEAN DIFFERENCE/ RATIO (95% CI) | P VALUE |
|---|---|---|---|---|
| **FACIAL SYMMETRY** | | | | |
| SURFACE MATCHING BETWEEN OS—MS (%) | 53.8 (9.6) | 50.7 (10.8) | -3.07 (-7.12 to 0.97) | .135 |
| AVERAGE DISTANCE BETWEEN OS—MS (MM) | 0.67 (0.16) | 0.74 (0.21) | 0.07(-0.01 to 0.14) | .074 |
| MAXIMUM DISTANCE BETWEEN OS—MS (MM) | 3.9 (1.01) | 4.0 (0.82) | 0.12 (-0.24 to 0.50) | .501 |
| **FACIAL WIDTHS** | | | | |
| FACIAL WIDTH (MM) | 118.2 (5.9) | 121.1 (7.2) | 2.87 (0.25 to 5.49) | .032* |
| GONION WIDTH (MM) | 116.5 (5.8) | 124.3 (7.5) | 7.86 (5.20 to 10.5) | .000*** |
| WIDTH RATIO | 1.02 (0.04) | 0.97 (0.04) | -0.04 (-0.06 to -0.02) | .000*** |
| **FACIAL HEIGHTS** | | | | |
| FACIAL HEIGHT (MM) | 111.3 (5.4) | 119.3 (5.7) | 8.05 (5.85 to 10.26) | .000*** |
| MIDDLE FACIAL HEIGHT (MM) | 66.0 (3.9) | 69.0 (4.3) | 3.04 (1.41 to 4.67) | .000*** |
| LOWER FACIAL HEIGHT (MM) | 49.7 (4.3) | 53.9 (4.3) | 4.23 (2.53 to 5.94) | .000*** |
| RATIO BETWEEN MIDDLE AND LOWER FACIAL HEIGHT | 1.34 (0.14) | 1.29 (0.14) | -0.05 (-0.11 to 0.01) | .077 |
| **FACIAL WIDTH-TO-HEIGHT RATIO** | 1.06 (0.06) | 1.02 (0.07) | -0.05 (-0.07 to -0.02) | .000*** |
| **FACIAL PROFILE** | | | | |
| FACIAL ANGLE (°) | 164.1 (4.9) | 162.9 (5.6) | -1.24 (-3.32 to 0.85) | .242 |
| ANGLE OF LOWER FACIAL HEIGHT (°) | 185.6 (8.7) | 186.4 (7.0) | 0.83 (-2.31 to 3.97) | .600 |
| FOREHEAD ANGLE (°) | 11.0 (4.6) | 11.5 (5.8) | 0.57 (-1.51 to 2.65) | .587 |
| GLABELLA PROMINENCE ANGLE (°) | 159.5 (5.1) | 156.5 (7.2) | -3.07 (-5.54 to -0.60) | .020* |
| **MOUTH** | | | | |
| MOUTH WIDTH (MM) | 45.4 (3.9) | 47.2 (3.2) | 1.78 (0.37 to 3.19) | .014* |
| UPPER VERMILION MIDDLE HEIGHT (MM) | 8.2 (1.1) | 8.9 (1.8) | 0.69 (0.09 to 1.29) | .024* |
| LOWER VERMILION MIDDLE HEIGHT (MM) | 9.5 (1.7) | 9.1 (1.8) | -0.39 (-1.09 to 0.31) | .270 |
| UPPER-LIP HEIGHT (MM) | 19.8 (2.4) | 21.7 (2.4) | 1.93 (0.85 to 2.85) | .000*** |
| RATIO BETWEEN UPPER LIP AND LOWER FACIAL HEIGHT | 0.40 (0.03) | 0.40 (0.03) | 0.00 (-0.01 to 0.02) | .477 |
| **NOSE** | | | | |
| NOSE WIDTH (MM) | 31.8 (2.2) | 34.8 (2.3) | 3.04 (2.14 to 3.93) | .000*** |
| NOSE HEIGHT (MM) | 43.0 (3.0) | 46.8 (3.4) | 3.77 (2.50 to 5.04) | .000*** |
| NASOLABIAL ANGLE (°) | 114.2 (10.3) | 112.5 (11.2) | -1.75 (-6.02 to 2.52) | .418 |
| **EYES** | | | | |
| INTER EYE DISTANCE (MM) | 34.4 (3.4) | 34.9 (3.2) | 0.41 (-0.90 to 1.73) | .535 |
| LEFT PALPEBRAL FISSURE (MM) | 27.7 (2.2) | 28.2 (2.3) | 0.54 (-0.36 to 1.45) | .236 |
| RIGHT PALPEBRAL FISSURE (MM) | 27.3 (2.0) | 27.8 (2.2) | 0.47 (-0.36 to 1.31) | .263 |

* *p* < .05;

** *p* < .01;

*** *p* < .001

OS—original facial shell; MS—mirrored facial shell; SD—standard deviation; CI—confidence interval; n—number of subjects.

younger ones. In the older group the differences between the male and female face widths were greater: the upper facial width was 8 mm wider and the lower facial width was 13 mm wider in the males (Table 3). The women had an increased ratio between the upper and lower facial widths. With a higher BMI, the ratio between the upper and lower facial widths decreased in terms of statistical significance in both genders. Age and body height had no influence on the ratio (Table 4).

Table 3. Descriptive statistics and comparison of the facial parameters between genders (independent samples *t*-test) in the older group (49 females, 41 males).

| PARAMETERS | FEMALES MEAN (SD) | MALES MEAN (SD) | MEAN DIFFERENCE/ RATIO (95% CI) | *P* VALUE |
|---|---|---|---|---|
| **FACIAL SYMMETRY** | | | | |
| SURFACE MATCHING BETWEEN OS—MS (%) | 42.1 (8.8) | 39.3 (8.6) | -2.72 (-6.19 to 0.74) | .122 |
| AVERAGE DISTANCE BETWEEN OS—MS (MM) | 0.85 (0.20) | 0.96 (0.22) | 0.11 (0.02 to 0.19) | .014* |
| MAXIMUM DISTANCE BETWEEN OS—MS (MM) | 4.2 (0.71) | 4.6 (0.59) | 0.42 (0.16 to 0.68) | .002** |
| **FACIAL WIDTHS** | | | | |
| FACIAL WIDTH (MM) | 124.4 (6.8) | 131.0 (6.5) | 6.59 (3.94 to 9.24) | .000*** |
| GONION WIDTH (MM) | 127.4 (8.4) | 138.7 (9.9) | 11.32 (7.67 to 14.98) | .000*** |
| WIDTH RATIO | 0.98 (0.05) | 0.95 (0.06) | -0.03(-0.05 to -0.01) | .006** |
| **FACIAL HEIGHTS** | | | | |
| FACIAL HEIGHT (MM) | 112.0 (6.9) | 123.8 (6.4) | 11.81 (9.16 to 14.46) | .000*** |
| MIDDLE FACIAL HEIGHT (MM) | 67.5 (4.5) | 73.5 (4.6) | 5.97 (4.09 to 7.85) | .000*** |
| LOWER FACIAL HEIGHT (MM) | 50.9 (5.2) | 55.4 (4.0) | 4.53 (2.67 to 6.38) | .000*** |
| RATIO BETWEEN MIDDLE AND LOWER FACIAL HEIGHT | 1.34 (0.16) | 1.33 (0.12) | -0.01 (-0.06 to 0.05) | .779 |
| **FACIAL WIDTH-TO-HEIGHT RATIO** | 1.11 (0.08) | 1.06 (0.06) | -0.05 (-0.08 to -0.03) | .000*** |
| **FACIAL PROFILE** | | | | |
| FACIAL ANGLE (°) | 172.9 (6.9) | 172.9 (6.7) | 0.03 (-2.68 to 2.73) | .985 |
| ANGLE OF LOWER FACIAL HEIGHT (°) | 193.0 (15.0) | 194.4 (15.9) | 1.41 (-4.70to 7.53) | .647 |
| FOREHEAD ANGLE (°) | 14.4 (7.5) | 24.7 (19.0) | 10.28 (4.52 to 16.04) | .001** |
| GLABELLA PROMINENCE ANGLE (°) | 162.0 (7.7) | 152.4 (6.3) | -9.66 (-12.47 to -6.86) | .000*** |
| **MOUTH** | | | | |
| MOUTH WIDTH (MM) | 44.3 (5.7) | 47.8 (5.5) | 3.48 (1.26 to 5.71) | .002** |
| UPPER VERMILION MIDDLE HEIGHT (MM) | 5.2 (1.6) | 5.3 (1.8) | 0.09 (-0.58 to 0.75) | .795 |
| LOWER VERMILION MIDDLE HEIGHT (MM) | 6.2 (1.9) | 6.4 (2.3) | 0.28 (-0.55 to 1.10) | .510 |
| UPPER-LIP HEIGHT (MM) | 20.1 (2.9) | 22.9 (2.8) | 2.85 (1.72 to 3.98) | .000*** |
| RATIO BETWEEN UPPER LIP AND LOWER FACIAL HEIGHT | 0.40 (0.04) | 0.41 (0.04) | 0.02 (-0.00 to 0.04) | .025* |
| **NOSE** | | | | |
| NOSE WIDTH (MM) | 35.3 (2.2) | 39.2 (3.9) | 4.86 (3.28 to 6.43) | .000*** |
| NOSE HEIGHT (MM) | 45.6 (3.8) | 50.4 (4.1) | 4.86 (3.28 to 6.43) | .000*** |
| NASOLABIAL ANGLE (°) | 110.1 (13.9) | 111.6 (14.5) | 1.50 (-4.14 to 7.14) | .598 |
| **EYES** | | | | |
| INTER EYE DISTANCE (MM) | 37.1 (3.7) | 39.5 (3.3) | 2.41 (1.04 to 3.79) | .001** |
| LEFT PALPEBRAL FISSURE (MM) | 26.0 (3.2) | 26.4 (3.2) | 0.47 (-0.80 to 1.75) | .465 |
| RIGHT PALPEBRAL FISSURE (MM) | 25.8 (3.1) | 26.7 (3.6) | 0.86 (-0.48 to 2.21) | .206 |

\* *p* < .05;

\*\* *p* < .01;

\*\*\* *p* < .001

OS—original facial shell; MS—mirrored facial shell; SD—standard deviation; CI—confidence interval; n—number of subjects.

## Facial heights

Our study revealed that male faces were longer than the female faces, with the differences being statistically significant. In addition to the total facial height, the middle and lower facial heights were also greater in the males (Table 2). Body height had a positive impact on facial height (p = 0.034), but BMI had no influence. Men with the same body height as women had statistically significant longer faces (Table 4). With age the total face height increased, because

**Table 4. Multiple-linear-regression model to evaluate the influence of sex, age, BMI and body height on the facial parameters presented with coefficient and *p*—value in round bracket.** Sex (male = 0, female = 1); age (younger group = 0, older group = 1), body height (meters).

| PARAMETERS | | | INDEPENEDENT VARIABLES | | |
|---|---|---|---|---|---|
| | Constant | Sex | Age | BMI | Body height |
| Surface matching between OS—MS (%) | 49.13 (.008)** | 3.00 (.100) | -11.23(.000)*** | -0.03 (.870) | 1.33(.893) |
| Average distance between OS—MS (mm) | 0.90(.021)* | -0.10 (.011)* | 0.21 (.000)*** | -0.00 (.442) | -0.04 (.837) |
| Maximum distance between OS—MS (mm) | 3.63 (.020)* | -0.24 (.115) | 0.61 (.000)*** | -0.03(.139) | 0.56 (.499) |
| Face width (mm) | 88.46 (.000)*** | -2.54 (.040)* | 6.94 (.000)*** | 0.50 (.000)*** | 12.19 (.071) |
| Gonion width (mm) | 70.94 (.000)*** | -5.97 (.000)*** | 7.84 (.000)*** | 1.28 (.000)*** | 13.76 (.054) |
| Width ratio | 1.13 (.000)*** | 0.03 (.007)** | -0.01 (.562) | -0.01 (.000)*** | -0.01 (.781) |
| Facial height (mm) | 90.04 (.000)*** | -7.99 (.000)*** | 3.20 (.012)* | 0.17 (.209) | 14.51 (.024)* |
| Middle facial height (mm) | 47.00 (.000)*** | -3.00 (.000)*** | 3.99(.000)*** | 0.04 (.714) | 12.05 (.010)* |
| Lower facial height (mm) | 47.47 (.000)*** | -4.01 (.000)*** | 1.62 (.081) | 0.03 (.792) | 3.27 (.482) |
| Ratio between middle and lower facial height | 1.00 (.000)*** | 0.05 (.060) | 0.04 (.188) | 0.00 (.911) | 0.17 (.250) |
| Facial width to-height-ratio | 0.99 (.000)*** | 0.05 (.000)*** | 0.03 (.022)* | 0.00(.058) | -0.02 (.775) |
| Facial angle (°) | 157.91 (.000)*** | 1.08 (.341) | 7.45 (.000)*** | 0.34 (.008)** | -1.45 (.816) |
| Angle of lower facial height (°) | 187.94 (.000)*** | -0.28 (.904) | -8.15 (.001)** | -.06 (.829) | -9.48 (.454) |
| Forehead angle (°) | 25.67 (.007)** | -8.15 (.000)*** | 5.79 (.012)* | 0.01 (.978) | -4.12 (.037)* |
| Glabella prominence angle (°) | 139.64 (.000)*** | 7.19 (.000)*** | 0.54 (.701) | -0.06 (.690) | 9.10 (.203) |
| Mouth width (mm) | 18.89 (.035)* | -0.69 (.429) | 0.44 (.643) | 0.13 (.195) | 14.12 (.004)** |
| Upper vermilion middle height (mm) | 8.38 (.008)** | -0.38 (.216) | -3.40 (.000)*** | 0.02 (.519) | -0.05 (.975) |
| Lower vermilion middle height (mm) | 6.96 (.061) | 0.25 (.500) | -2.71 (.000)*** | -0.04 (.312) | 1.81 (.364) |
| Upper lip height (mm) | 20.73 (.000)*** | -2.37 (.000)*** | 1.18 (.031)* | -0.04 (.442) | 1.21 (.659) |
| Ratio between upper lip lower facial height | 0.43 (.000)*** | -0.01 (.045)* | 0.01 (.196) | -0.00 (.201) | -0.00 (.943) |
| Nose width (mm) | 19.73 (.000)*** | -2.42 (.000)*** | 3.39 (.000)*** | 0.22 (.000)*** | 5.69 (.039)* |
| Nose height (mm) | 29.75 (.000)*** | -3.16 (.000)*** | 3.43 (.000)*** | 0.10 (.213) | 8.30 (.026)* |
| Nasolabial angle (°) | 117.43 (.000)*** | -0.28 (.905) | 0.92 (.717) | -0.59 (.027)* | 5.19 (.685) |
| Inter-eye distance (mm) | 20.56 (.002)** | -0.52 (.423) | 4.10 (.000)*** | 0.07 (.349) | 7.28 (.040)* |
| Left palpebral fissure (mm) | 23.18 (.000)*** | -0.23 (.661) | -2.37 (.000)*** | 0.16 (.009)** | 0.83 (.769) |
| Right palpebral fissure (mm) | 20.76 (.000)*** | -0.19 (.715) | -1.81 (.002)** | 0.15 (.012)* | 1.99 (.492) |

\* *p* < .05;

\*\* *p* < .01;

\*\*\* *p* < .001

of the increasing of the middle facial height. Age and body height had no impact on the lower facial height (Table 4). However, the females' lower facial heights were on average 4 mm less than the males. The ratio between the middle and lower facial height was larger in women, because of the men's larger lower facial height (Table 2). Age, BMI and body height had no impact on the lower facial height.

## The ratio between facial width and height

The women's width-to-height ratio was larger than the men's, which means their faces were rounder (Table 2). Body height had no impact on the ratio, but the BMI did. The men and women with a larger BMI had a statistically significant larger facial ratio, but this was clinically irrelevant (a 5-unit-larger BMI means a 0.015 higher ratio). With age the ratio increased (Table 4).

## Facial profile

There were no differences in facial profiles between the men and women. The facial height had no impact on the facial angle, but the BMI did. People with a larger BMI had a more concave facial profile (Table 4). Despite the statistically significant difference, the clinical correlation was irrelevant, due to the small difference. A 1-unit-larger BMI means a 0.4˚ larger facial angle, and this cannot be described as a visible change.

With age the facial angle was more obtuse, which means a more concave facial profile. The angle of the lower facial height was the same in the women and the men. Body height and BMI had no influence on the angle of the lower facial height. With age the angle was smaller, which means a more intruded lip part (Table 4).

There was no difference in forehead inclination between the younger men and women (Table 2). Body height and BMI had no impact on the forehead angle. With age the forehead angle increased for the men and women, which means a larger forehead inclination. In the older group the men had a much larger forehead inclination, which means more prominent supraorbital arches. The differences were statistically significant for the men and women (Table 3).

The glabella's prominence angle was larger in the women than in the men. With age, the difference between the sexes increased. Body height and BMI had no impact on the glabella's prominence angle.

## Mouth

The mouth was, on average, 2 mm wider in the men than in the women (Table 2). Mouth width increased with increasing body height in the men and women (10-cm-taller males/females have 1.5-mm-wider mouth) (Table 4). In contrast, BMI and age had no impact on mouth width. In the younger males the upper and lower vermilion middle heights were almost the same, but in the younger females the lower vermilion middle height was larger than the upper, due to the more pronounced Cupid's bow. BMI and body height had no impact on the upper and lower vermilion middle heights. With age, the upper and lower vermilion middle height decreased in the males and females. Upper-lip height was greater in the men, and this increased with age. Facial height and BMI did not influence the upper-lip height. The ratio between the upper-lip and lower facial heights was the same in both sexes. There was no change in the ratio with aging.

## Nose

The men had longer and wider noses than the women. The taller men and women had longer and wider noses (Table 4). Nose height and width increased with aging. BMI had no impact on nose height, but influenced nose width, as men and women with higher BMIs had wider noses. There were no differences between the men and women in the nasolabial angle, but there was an impact of BMI on the nasolabial angle. People with larger BMIs had smaller angles (Table 4). Facial height and age did not influence the angle.

## Eyes

There was no difference in eye-gap width between the men and women (Table 2). With increased BMI the palpebral fissure width increased, and the differences were statistically significant. The eye gap was smaller in the older group (Table 3). The inter-eye distance was the same in the women and the men, but this increased with age. There was a positive correlation

between body height and inter-eye distance, but the BMI had no impact on the inter-eye distance (Table 4).

## Discussion

In our study facial sexual dimorphism was evaluated in both the younger and older adults. Although gender-dependent facial characteristics in younger adults and growing faces were already observed [11, 12], there is a shortage of studies evaluating the effect of aging on facial sexual dimorphism.

The sample size in our study was large enough to eliminate the natural differences in facial shape. We divided the subjects into younger and older adults to study the effect of aging on facial sexual dimorphism. The strength of this study is that factors such as age, BMI and body height are considered, which provided a clear picture of the distinctive facial gender features.

The method used in our study is a well-established, non-invasive, reproducible and accurate method [11]. To achieve the accuracy and reproducibility of the scans, our subjects were seated with a natural head position [12]. Most of the parameters used in our study have been previously used in 3D cephalometric studies. The facial symmetry was evaluated with an established method [13] that takes into account all the facial points and allows for a full face analysis [14]. To eliminate the size-related changes such as body mass and height a linear regression model was used.

Studies have shown that averageness, symmetry and sexual dimorphism are the main factors for the biologically based standards of beauty [2]. It is well known that no human face is perfectly symmetric, as there are always areas of asymmetry between the left and right-hand sides of the face that are considered to be physiological [15]. In our study the male faces were more asymmetric than the female faces, but the result was statistically significant only for the subjects in the older group. Less symmetric male faces have also been shown in adolescents [13]. The more symmetric female faces is in agreement with findings that symmetry is relatively more important for the beauty appreciation of female faces than male faces [16]. With aging face asymmetry becomes more evident, probably due to the superficial textural wrinkling of the skin and changes in the three-dimensional (3D) topography of the underlying structures, both the soft-tissue envelope and the underlying facial skeleton [17].

The faces were wider for the men than the women, as has been described before [18]. Differences in the facial widths between the genders were greater in the older adults, independent of the BMI. Wide jaws in men are attractive to women [7] and in our study we found that the men had wider jaws than the women. In our study larger ratio between upper and lower facial width in women, manifests clinically as triangular faces. In contrast, the men's ratio between the upper and lower facial width was smaller, resulting in a squarer face. We found that males also have longer faces, consistent with a published study [19]. Importantly, men with the same body height as women had longer faces; this has not been described previously.

We demonstrated that the females have a larger width-to-height ratio than the males, in agreement with a Turkish study [20], which means that the shape of the female face was rounder, while the men's faces were more oval. This is in contrast to the result of an anthropometric study, which did not find sexual dimorphism in the width-to-height ratio [21], but most probably due to an ethnically conditioned face. A study in 1000 Japanese adults has shown that the predominant facial shape variation is in the height-to-weight proportion, but found no differences between genders [22]. With age the female facial form became rounder, but the male form varied from oval to rectangular.

Surprisingly, there were no differences between men's and women's of facial profiles. We expected to find a larger facial angle in men, which means more concave facial profile.. Our result could be a consequence of the larger chins in the Slovenian female population compared with

other ethnic groups of Caucasian ancestry [23]. With age the facial profile becomes more concave, which can be explained by a lengthening of the lower facial height and pogonion repositioning.

In contrast to the general belief that men typically have more pronounced brow ridges [24], we found no differences between the younger men and women for the forehead angle. Only the older males had a larger forehead inclination and more prominent supraorbital arches than the older women in our study. To more accurately evaluate the shape of the glabella region and the lower forehead we measured the glabella's prominence angle. As expected, women had a more obtuse angle, meaning a flatter forehead. In men, the glabella prominence was more protruded and is connected with the more prominent supraorbital ridges in men [25]. Age did not influence the glabella prominence angle.

Both the forehead angle and the glabella prominence angle involved the trichion point, which is determined as the point between the forehead and the scalp. In older adults, special attention must be addressed when determining the point due to hair loss.

Women have a smaller mouth with fuller (larger) lips, which is considered to be more attractive [26]. The upper and lower vermilions in the younger males were of the same medial height. In females the lower vermilion was higher than the upper, probably because we were measuring the upper medial vermilion height, which is the lowest part of the Cupid's bow. Our study established that both of the vermilions become much narrower with age, most probably due to the loss of supportive tissue and gravity [27, 28]. The upper lip was longer in males, which became even more pronounced with aging.

The nose is a very significant part of the face and has its own characteristics. The men's noses were longer and wider, which has been observed before [18]. In our study we confirmed the nose lengthening and extension with age, consistent with published data [29]. The lengthening of the nose is a consequence of the intrinsic loosening of the lower lateral alar cartilages and the supporting ligaments [30].

One of the main characteristic of female beauty is large eyes [31]. Surprisingly, we found no difference in the palpebral fissure width between men and women. With age the gap gets smaller, as a consequence of senile ptosis of the upper eyelid. The inter-eye distance was the same in women as in men, but this increases with age.

Our study confirmed that older adults have a significantly higher BMI than younger adults. It is known that an increased BMI has a larger influence on the transverse dimensions of the face [32]. Our study confirmed not only wider faces with increasing BMI, but also longer facial widths with aging, independently of the BMI. With a higher BMI, the ratio between the upper and lower facial width becomes smaller [33, 34], highlighting the impact of body weight on the lower facial width, which is also a characteristic of aging. We confirmed previous findings about the influence of BMI on the facial ratio [34, 35].

As expected, body height influences the facial dimensions. There are, for example, some studies predicting body height from the head and face dimensions [36]. In our study taller men and women have longer faces, but also longer and wider noses. The distance between the eyes is greater and the mouth gap is wider.

The face is one of the most diverse parts of the human body. In today's society, which is dictated by a general social acceptance and the associated aesthetics, the appearance of the face has an important role. It has been suggested that sexual dimorphism and symmetry in faces are signals advertising mate quality by providing evidence that there must be a biological mechanism linking the two traits during development [37]. Facial attractiveness as symmetry, averageness and sexual dimorphism have been suggested to provide signals of biological quality, especially health. There are data that indicate the weak links between attractive facial traits and health [38], but also studies that indicate the appeal of averageness and femininity in female faces and masculinity in male faces [39].

Sexual dimorphism has an important impact on evolutionary and anthropometric explanations of social interactions. Sexual dimorphism together with averageness influences facial attractiveness [40]; but these subjects are beyond the scope of our study. Our results contribute to a quantitative evaluation of facial morphology, which is essential for surgeons when planning facial surgical procedures. Differences in males and females have a practical importance in several areas of surgery, such as craniofacial, maxillofacial and plastic surgery, not only for feminization procedures but also when the main task is to reproduce the anatomical structures to a specific biological profile. Rejuvenation procedures and other major facial reconstructions should be performed with an understanding of specific morphologic facial characteristics.

The strenght of our study is that facial sexual dimorphism was confirmed not only in young adults, but also in older adults. Moreover, we found that all gender-dependent characteristics were more pronounced with aging. We confirmed known differences in several facial characteristics, but our results relating to longer faces in men than in women with the same body height have not been described before. In addition, we found facial widening with age, despite an unchanged BMI, and confirmed a more pronounced lower forehead in males of all ages.

## Conclusions

We demonstrated facial sexual dimorphism, including shape, form and facial ratios in younger and older adults. The differences are more pronounced in the older adults, especially in terms of male facial asymmetry. The appreciation of facial characteristics is important for rejuvenation and aesthetic surgery, but also for craniofacial surgery, especially in orthognathic, syndromic patients and feminization procedures, because gender differences have an important impact on planned facial appearance.

## Supporting information

**S1 Dataset. Coordinates of all reference points exported from the software package Rapidform®2006 (Inus Technology Inc., Seoul, Korea) to excel worksheet.**
(XLSX)

**S2 Dataset. Descriptive statistics and calculated observed parameter values of all the subjects for final statistical analyses and comparison.**
(SAV)

## Author Contributions

**Conceptualization:** Zala Skomina, Nataša Ihan Hren.

**Data curation:** Zala Skomina.

**Formal analysis:** Zala Skomina, Miha Verdenik.

**Investigation:** Zala Skomina.

**Methodology:** Zala Skomina, Miha Verdenik, Nataša Ihan Hren.

**Project administration:** Nataša Ihan Hren.

**Software:** Miha Verdenik.

**Supervision:** Nataša Ihan Hren.

**Validation:** Zala Skomina.

**Visualization:** Zala Skomina, Miha Verdenik.

Writing – **original draft:** Zala Skomina.

Writing – **review & editing:** Miha Verdenik, Nataša Ihan Hren.

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
