## [Decision Letter · Decision Letter 0]

3 Jan 2020

PONE-D-19-35009

The Effect of Ageing and Body Characteristics on Facial Sexual Dimorphism in Caucasian Population

PLOS ONE

Dear Mrs. Skomina,

Thank you for submitting your manuscript to PLOS ONE. After careful consideration, we feel that it has merit but does not fully meet PLOS ONE’s publication criteria as it currently stands. Therefore, we invite you to submit a revised version of the manuscript that addresses the points raised during the review process.

We would appreciate receiving your revised manuscript by Feb 17 2020 11:59PM. To enhance the reproducibility of your results, we recommend that if applicable you deposit your laboratory protocols in protocols.io, where a protocol can be assigned its own identifier (DOI) such that it can be cited independently in the future. For instructions see: http://journals.plos.org/plosone/s/submission-guidelines#loc-laboratory-protocols

We look forward to receiving your revised manuscript.

Kind regards,

Mateusz Koziej, MD, PhD

Academic Editor

PLOS ONE

Journal Requirements:

Reviewers' comments:

Reviewer's Responses to Questions

**Comments to the Author**

1. Is the manuscript technically sound, and do the data support the conclusions?

Reviewer #1: Partly

Reviewer #2: Yes

Reviewer #3: Yes

2. Has the statistical analysis been performed appropriately and rigorously? 

Reviewer #1: Yes

Reviewer #2: I Don't Know

Reviewer #3: Yes

3. Have the authors made all data underlying the findings in their manuscript fully available?

Reviewer #1: Yes

Reviewer #2: Yes

Reviewer #3: Yes

4. Is the manuscript presented in an intelligible fashion and written in standard English?

Reviewer #1: Yes

Reviewer #2: Yes

Reviewer #3: Yes

5. Review Comments to the Author

Reviewer #1: This manuscript is interesting, it includes big cohort of examined patients and its study design is well prepared. However before accepting its publication I think authors need to perform manuscript revision.

1. In the introduction section the sentence: “The loss of supporting bones and teeth strongly influences the covering soft tissues and leads to the formation of wrinkles and compensatory mechanisms of the mimic muscles…” needs revision, in its current form one can conclude that wrinkles are formatted because of the loss of bones. The are of course some changes in the status of elastin and collagen fibers.

2. Authors should add information what was the distance between position of scanner and examined person.

3. In the discussion section in the sentence “Differences in facial widths between male

and female were greater in older adults, which means that gender dimorphism gets more

evident later in life...” the conclusion in my opinion is drawn to early as the body weight gets

bigger in older patients the difference in facial width is connected to increased BMI rather

than gender dimorphism. It would be ok to write such a conclusion if older patients would

have similar BMI to young adults.

4. Later in discussion the sentence “In Czech boys and girls at the age of 12 to 13 years there were no significant gender specific differences in facial shape, but from the age of 14 the differences were apparent [27].” is not needed. It do not put any relevant data to the text, here authors write about young and older adults not about children…

5. I suggest to improve the figure 2, the line n-prn should be more in the middle especially in the picture where eyes points are shown, the distance between enL-enR and the line n-prn should be almost equal…

Reviewer #2: Overall comments

Study is interesting.

The study group is too small (especially for analysis of so many different factors). I recommend to include more patients.

Minor grammar corrections should be performed.

Introduction

Well-structured. Minor grammar corrections should be made.

MM

No data about enrollment of the patients (who performed? where? how they have been approached, etc. ).

No data about excluded (if any?) patients.

No data about statistical analysis of data with non-Gaussian distribution. It is not possible to fully asses statistical methods used by authors.

How many facial scans per person have been performed? How many of them have been repeated? How the same position of the patients have been controlled?

Clearer and better quality figures should be provided.

Discussion

There is too many repetitions from results. More other studies should be discussed and cited to give the reader comperhensive evaluation of the topic.

Reviewer #3: Congratulations on a rigorous and thoughtful paper. I enjoyed reading it.

Several questions:

1. I agree with your method of calculating symmetry using the best fit approach between native and reflective faces. Using manually selected points to generate a plane of symmetry introduces far too much error. For your anthropometric landmarks however a single examiner selected the spots manually. Do you have any information on how reproducible this process is--in other words any intra-observer reliability data?

2. One of the most powerful components of facial feminization surgery involves reducing the prominent supraorbital bandeau that juts out in many men from the forehead, over the top of the frontal sinus. The forehead profile is more S shaped than C shaped. While you nicely describe the forehead angle, is there a way of capturing the forehead shape differences with the prominent bandeau shape seen more in male and females using your data?

3. Do you think the increase in forehead angle with age could just be a result of the trichion moving further back with hair loss? It is hard to understand how this could happen otherwise?

4. Some minor editorial suggestions :

A) In the abstract results section instead of "was more round and their faces were smaller, taking account their body characteristics" you could say " was more round and their faces were smaller, after normalizing for body size."

B) Introduction is misspelled (Intruduction should be Introduction)

C) Sentence 4 of the introduction should start with the pronon The "The majority of ….

D) Sentence 2 of paragraph 3 of Intro shoudl also start with The " The face is changing trough" and Through is misspelled

E) The last sentence of paragraph 3 of Introduction should say "mimetic muscles" not "mimic muscles"

F) Perhaps instead of "Eyes" consider "palpebral fissures" as it is the fissure rather than the globe that you are measuring

6. PLOS authors have the option to publish the peer review history of their article (what does this mean?). If published, this will include your full peer review and any attached files.

Reviewer #1: No

Reviewer #2: No

Reviewer #3: Yes: Helena O. Taylor

---

## [Author Response · Author response to Decision Letter 0]

12 Feb 2020

Our answers to Reviewers:

Reviewer #1: 

This manuscript is interesting, it includes big cohort of examined patients and its study design is well prepared. However before accepting its publication I think authors need to perform manuscript revision.

1. In the introduction section the sentence: “The loss of supporting bones and teeth strongly influences the covering soft tissues and leads to the formation of wrinkles and compensatory mechanisms of the mimic muscles…” needs revision, in its current form one can conclude that wrinkles are formatted because of the loss of bones. There are of course some changes in the status of elastin and collagen fibers.

We agree with the reviewer, so the sentence in Paragraph 3 in the Introduction section has been revised as follows: 

Facial aging results from a combination of changes in soft tissue (such as changes in the status of elastin and collagen fibers), with bone loss in specific areas of the facial skeleton contributing to the features of the aging [10].

2. Authors should add information what was the distance between position of scanner and examined person.

The distance between the examined person and the scanner was 50–70 cm. This information was added in Paragraph 5 of the Materials and Methods section.

3. In the discussion section in the sentence “Differences in facial widths between male

and female were greater in older adults, which means that gender dimorphism gets more

evident later in life...” the conclusion in my opinion is drawn to early as the body weight gets

bigger in older patients the difference in facial width is connected to increased BMI rather

than gender dimorphism. It would be ok to write such a conclusion if older patients would

have similar BMI to young adults.

Our data revealed differences in both transversal facial parameters between the sexes in younger and older adults and we have shown that the differences were greater in the older group. The reviewer is correct that an increased BMI impacts the transverse dimensions of the face. However, using the multiple linear regression model (Table 4) we demonstrated a significantly shorter facial and gonial width in women, independently of the BMI.

Based on the comment in Paragraph 5 in the Discussion, the sentence was revised: 

Differences in the facial widths between the genders were greater in the older adults, independent of the BMI. In the Discussion section in Paragraph 12 the influence of the body-mass index on transverse parameters was outlined: 

Our study confirmed that older adults have a significantly higher BMIs than younger adults. It is known that an increased BMI has a larger influence on the transverse dimensions of the face. Our study confirmed not only wider faces with increasing BMI, but also longer facial widths with aging, independently of the BMI. 

4. Later in discussion the sentence “In Czech boys and girls at the age of 12 to 13 years there were no significant gender specific differences in facial shape, but from the age of 14 the differences were apparent [27].” is not needed. It do not put any relevant data to the text, here authors write about young and older adults not about children…

We agree with the reviewer and the sentence and the reference were removed from the manuscript.

5. I suggest to improve the figure 2, the line n-prn should be more in the middle especially in the picture where eyes points are shown, the distance between enL-enR and the line n-prn should be almost equal…

Figure 2 has been improved as suggested. The line n-prn was moved to the middle so the distance between the n-prn line and the enL point is equal to the distance between the line and the enR point. All the figures have been improved and corrected with the PACE digital diagnostic tool to meet the PLOS requirements for figure quality and format.

Reviewer #2:

Overall comments

Study is interesting.

The study group is too small (especially for analysis of so many different factors). I recommend to include more patients.

The sample number was calculated with a sample-size calculation for a predicted study power of 0.8 in comparing two independent samples tested for numerical variables by a professional biostatistician. The hypothesis was that the difference in facial height between younger women and men is significant and it was the basis for our sample number determination. However, we have added 20 more subjects in the revised manuscript, which were already collected in the sampling phase of the study. The total number of subjects is now 200. Due to the increased number of subjects, all the tables were revised, but the interpretation of the results has not changed.

Minor grammar corrections should be performed.

The manuscript was edited by Dr Paul McGuiness, a professional proofreader and a native speaker of English.

Introduction

Well-structured. Minor grammar corrections should be made.

Grammar corrections were made.

MM

No data about enrolment of the patients (who performed? where? how they have been approached, etc…).

No data about excluded (if any?) patients.

We now provide a description of the patient enrolment in the Materials and Methods section in Paragraph 1: 

The younger adults were students at the School of Medicine of Ljubljana, Slovenia. The older group contained residents of five retirement homes in Ljubljana, Slovenia. 

We have previously provided exclusion factors in the Materials and Methods section in Paragraph 1: 

The exclusion criteria were a craniofacial anomaly, a history of major facial trauma, or orthognathic surgery, facial paresis and tremor. Male subjects with facial hair were also excluded.

No data about statistical analysis of data with non-Gaussian distribution. It is not possible to fully asses statistical methods used by authors.

Our statistical methods were chosen and performed by a professional biostatistician. Due to the large sample size, normality tests are not needed. However, we analyzed the Q-Q Plots and confirmed that all the observed parameters were normally distributed for all the groups and even more importantly we checked for the homogeneity of variance, as assessed by Levene'.

How many facial scans per person have been performed? How many of them have been repeated? How the same position of the patients have been controlled?

Based on this comment we now provide a better description of the patient position and its control in the Material and Methods section in Paragraph 4: 

The natural head position was achieved after instructions and exercises by moving the head up and down a few times and then stopping the movement and looking into the distance. A relaxed, closed-mouth position was achieved with a repeated wide opening and closing the mouth until light contact of the lips was achieved. A single facial scan required less than 10 seconds, so subjects were able to maintain their position. 

After scanning the quality of each scan was checked and 5% of the scans needed to be repeated.

Clearer and better quality figures should be provided.

All the figures have been improved and corrected with the PACE digital diagnostic tool to meet the PLOS requirements for figure quality and format.

Discussion

There is too many repetitions from results. More other studies should be discussed and cited to give the reader comperhensive evaluation of the topic.

We revised the Discussion as suggested by the reviewer. The repetitions from the results were removed or changed. Paragraphs 12 and 13 were added to emphasize the body-mass index and body height impact on facial characteristics. A few more comparable studies were included in the Discussion section.

Reviewer #3: 

Congratulations on a rigorous and thoughtful paper. I enjoyed reading it.

Several questions:

1. I agree with your method of calculating symmetry using the best fit approach between native and reflective faces. Using manually selected points to generate a plane of symmetry introduces far too much error. For your anthropometric landmarks however a single examiner selected the spots manually. Do you have any information on how reproducible this process is--in other words any intra-observer reliability data?

All the measurements were made by one rater. Before the study, the intra-rater reliability was verified with intra-class correlation, which is the method for numerical variables (equals Cohen’s Kappa coefficient in categorical variables). The variation in the choice of the exact tracing points was measured on 10 randomly chosen facial scans, on which cephalometric points were placed twice. The computed ICC values are provided in the attached table. The statistically significant results also confirm that the method was reliable and that it does not introduce any bias. The short description was added to the Materials and Methods section in Paragraph 6: 

Before the study, the intra-rater reliability was verified with an intraclass correlation and we confirmed that the method is reliable and that it does not introduce any bias.

Table. The repeatability of the method is presented as a comparison of two measurements on the same scan (N – number of scans, ICC – intraclass correlation coefficient).

Cephalometric parameter N First measurement Second measurement ICC

Facial width (mm) 10 120.77 122.61 0.906

Mouth width (mm) 10 43.24 46.75 0.928

Facial height (mm) 10 119.04 119.36 0.991

Middle facial height (mm) 10 70.64 70.78 0.735

Lower facial height (mm) 10 57.52 57.45 0.859

Upper-lip height (mm) 10 22.04 21.27 0.905

Upper vermilion middle height (mm) 10 9.73 9.42 0.528

Lower vermilion middle height (mm) 10 9.74 9.46 0.641

Nose height (mm) 10 46.81 46.11 0.974

Gonion width (mm) 10 122.55 124.31 0.953

Nose width (mm) 10 34.22 35.12 0.986

Facial angle (°) 10 163.79 164.31 0.966

Forehead angle (°) 10 15.68 18.48 0.846

Angle of lower facial height (°) 10 177.35 179.14 0.895

Nasolabial angle (°) 10 110.61 110.25 0.921

Inter-eye distance (mm) 10 33.73 34.01 0.993

Left palpebral fissure (mm) 10 27.12 27.33 0.921

Right palpebral fissure (mm) 10 27.07 27.60 0.895

2. One of the most powerful components of facial feminization surgery involves reducing the prominent supraorbital bandeau that juts out in many men from the forehead, over the top of the frontal sinus. The forehead profile is more S shaped than C shaped. While you nicely describe the forehead angle, is there a way of capturing the forehead shape differences with the prominent bandeau shape seen more in male and females using your data?

With our method the forehead shape differences between genders could be theoretically compared with the superposition of the forehead region, but the comparison of forehead regions of different subjects in our study would be very inaccurate and not an exact method, due to different facial sizes between sexes.

However, based on the comment of this reviewer we have added a new parameter, the glabella prominence angle, to more accurately describe the shape of the lower part of the forehead. The angle is between the nasion, glabella and trichion points. The glabella’s prominence angle described more prominent lower forehead in men and phenotypically is connected with more prominent supraorbital ridges in men. The description of this angle was added to the Materials and Methods section in Paragraph 14. 

We demonstrated that glabella prominence angle is significantly larger in women and this was added to the Results section in Paragraph 9 and in Tables 2, 3 and 4:

The glabella's prominence angle was larger in the women than in the men. With age, the difference between the sexes increased. Body height and BMI had no impact on the glabella’s prominence angle.

The interpretation of the glabella prominence differences was added to the Discussion section in Paragraph 8: 

To more accurately evaluate the shape of the glabella region and lower forehead we measured the glabella’s prominence angle. As expected women had a more obtuse angle, meaning a flatter forehead. In men, the glabella prominence was more protruded and is connected with more prominent supraorbital ridges in men [27]. Age did not influence the glabella’s prominence angle.

3. Do you think the increase in forehead angle with age could just be a result of the trichion moving further back with hair loss? It is hard to understand how this could happen otherwise?

The trichion point is determined as the point between the forehead and the scalp. In young subjects, the point is positioned on the hairline, but in older subjects, due to hair loss, the point is placed where forehead levels to the cranium curvature. Even if the trichion point is accurately determined, we are aware of possible errors. We have added this information to the Discussion section in Paragraph 8:

Both the forehead angle and glabella’s prominence angle involved the trichion point, which is determined as the point between the forehead and the scalp. In older adults, special attention must be addressed while determining the point due to hair loss.

4. Some minor editorial suggestions :

A) In the abstract results section instead of "was more round and their faces were smaller, taking account their body characteristics" you could say " was more round and their faces were smaller, after normalizing for body size."

B) Introduction is misspelled (Intruduction should be Introduction)

C) Sentence 4 of the introduction should start with the pronon The "The majority of ….

D) Sentence 2 of paragraph 3 of Intro shoudl also start with The " The face is changing trough" and Through is misspelled

E) The last sentence of paragraph 3 of Introduction should say "mimetic muscles" not "mimic muscles"

F) Perhaps instead of "Eyes" consider "palpebral fissures" as it is the fissure rather than the globe that you are measuring

All the suggestions were taken into account and are incorpora

---

## [Decision Letter · Decision Letter 1]

24 Mar 2020

PONE-D-19-35009R1

Effect of Ageing and Body Characteristics on Facial Sexual Dimorphism in the Caucasian Population

PLOS ONE

Dear Mrs. Skomina,

Thank you for submitting your manuscript to PLOS ONE. After careful consideration, we feel that it has merit but does not fully meet PLOS ONE’s publication criteria as it currently stands. Therefore, we invite you to submit a revised version of the manuscript that addresses the points raised during the review process.

We would appreciate receiving your revised manuscript by May 08 2020 11:59PM. To enhance the reproducibility of your results, we recommend that if applicable you deposit your laboratory protocols in protocols.io, where a protocol can be assigned its own identifier (DOI) such that it can be cited independently in the future. For instructions see: http://journals.plos.org/plosone/s/submission-guidelines#loc-laboratory-protocols

We look forward to receiving your revised manuscript.

Kind regards,

Mateusz Koziej, MD, PhD

Academic Editor

PLOS ONE

Additional Editor Comments (if provided):

Authors has addressed all the mentioned issues.

Some changes need to be made before the manuscript can be accepted.

In table 1. Please provide quartiles Q1,Q3.

Material – change for “Study group”

Methods – change for Protocol

I would like to ask authors to include in statistical analysis section 1-2 sentences regarding how to interpret the MLR test. Just to make it clear for the reader. What significant coefficient in this analysis actually means.

Reviewers' comments:

Reviewer's Responses to Questions

**Comments to the Author**

1. If the authors have adequately addressed your comments raised in a previous round of review and you feel that this manuscript is now acceptable for publication, you may indicate that here to bypass the “Comments to the Author” section, enter your conflict of interest statement in the “Confidential to Editor” section, and submit your "Accept" recommendation.

Reviewer #1: All comments have been addressed

Reviewer #3: All comments have been addressed

2. Is the manuscript technically sound, and do the data support the conclusions?

Reviewer #1: Yes

Reviewer #3: (No Response)

3. Has the statistical analysis been performed appropriately and rigorously? 

Reviewer #1: Yes

Reviewer #3: I Don't Know

4. Have the authors made all data underlying the findings in their manuscript fully available?

Reviewer #1: Yes

Reviewer #3: Yes

5. Is the manuscript presented in an intelligible fashion and written in standard English?

Reviewer #1: Yes

Reviewer #3: Yes

6. Review Comments to the Author

Reviewer #1: (No Response)

Reviewer #3: The authors have provided thorough and thoughtful responses to reviewers and made the suggested explanations and corrections.

7. PLOS authors have the option to publish the peer review history of their article (what does this mean?). If published, this will include your full peer review and any attached files.

Reviewer #1: No

Reviewer #3: Yes: Helena O. taylor

---

## [Author Response · Author response to Decision Letter 1]

1 Apr 2020

Our Answers to Editor:

In table 1. Please provide quartiles Q1,Q3.

Quartiles Q1, Q3 were added in Table 1. 

Material – change for “Study group”

Material section was changed in Study group.

Methods – change for Protocol

Methods section was changed in Protocol section.

I would like to ask authors to include in statistical analysis section 1-2 sentences regarding how to interpret the MLR test. Just to make it clear for the reader. What significant coefficient in this analysis actually means.

Based on this comment we now provide a better interpretation of the multiple-linear-regression model and the meaning of signifiant coefficient in the Statistical analysis section:

The significant regression coefficient (marked with *; **; *** in Table 3) shows you how much the dependent variable is expected to change when that independent variable increases by one, holding all the other independent variables constant. For example; in older group face width is 7 mm longer than in younger group with unchanged BMI, body height and gender.

---

## [Editor Report · Decision Letter 2]

2 Apr 2020

PONE-D-19-35009R2

Effect of Ageing and Body Characteristics on Facial Sexual Dimorphism in the Caucasian Population

PLOS ONE

Dear Mrs. Skomina,

Thank you for submitting your manuscript to PLOS ONE. After careful consideration, we feel that it has merit but does not fully meet PLOS ONE’s publication criteria as it currently stands. Therefore, we invite you to submit a revised version of the manuscript that addresses the points raised during the review process.

We would appreciate receiving your revised manuscript by May 17 2020 11:59PM. To enhance the reproducibility of your results, we recommend that if applicable you deposit your laboratory protocols in protocols.io, where a protocol can be assigned its own identifier (DOI) such that it can be cited independently in the future. For instructions see: http://journals.plos.org/plosone/s/submission-guidelines#loc-laboratory-protocols

We look forward to receiving your revised manuscript.

Kind regards,

Mateusz Koziej, MD, PhD

Academic Editor

PLOS ONE

Additional Editor Comments (if provided):

Thank you for correction.

Please change sentence in statistical analysis as follows:

The significant regression coefficient (marked with *; **; ***) shows how the dependent variable is expected to change when that independent variable increases by one, holding all the other independent variables constant.

After that, the article will meet all criteria to be accepted in PLOS ONE.

---

## [Author Response · Author response to Decision Letter 2]

3 Apr 2020

Our Answers to Editor:

Please change sentence in statistical analysis as follows:

The significant regression coefficient (marked with *; **; ***) shows how the dependent variable is expected to change when that independent variable increases by one, holding all the other independent variables constant.

The sentense in statistical analysis was changed.

---

## [Editor Report · Decision Letter 3]

6 Apr 2020

Effect of Ageing and Body Characteristics on Facial Sexual Dimorphism in the Caucasian Population

PONE-D-19-35009R3

Dear Dr. Skomina,

We are pleased to inform you that your manuscript has been judged scientifically suitable for publication and will be formally accepted for publication once it complies with all outstanding technical requirements.

With kind regards,

Mateusz Koziej, MD, PhD

Academic Editor

PLOS ONE

---

## [Editor Report · Acceptance letter]

1 May 2020

PONE-D-19-35009R3 

Effect of Aging and Body Characteristics on Facial Sexual Dimorphism in the Caucasian Population 

Dear Dr. Skomina:

I am pleased to inform you that your manuscript has been deemed suitable for publication in PLOS ONE. Congratulations! Your manuscript is now with our production department. 

With kind regards,

on behalf of

Dr. Mateusz Koziej 

Academic Editor

PLOS ONE